# Negative Evidence Matters in Interpretable Histology Image Classification

**Soufiane Belharbi**[1]                                         soufiane.belharbi.1@ens.etsmtl.ca
**Marco Pedarsoli**[1]                                          marco.pedersoli@etsmtl.ca
**Ismail Ben Ayed**[1]                                          ismail.benayed@etsmtl.ca
**Luke McCaffrey**[2]                                           luke.mccaffrey@mcgill.ca
**Eric Granger**[1]                                             eric.granger@etsmtl.ca

[1] *Laboratoire d'imagerie, de vision et d'intelligence artificielle, Dept. of Systems Engineering, ETS Montreal, Canada*

[2] *Goodman Cancer Research Centre, Dept. of Oncology, McGill University, Montreal, Canada*

**Editors:** Under Review for MIDL 2022

## Abstract

Using only global image-class labels, weakly-supervised learning methods, such as class activation mapping, allow training CNNs to jointly classify an image, and locate regions of interest associated with the predicted class. However, without any guidance at the pixel level, such methods may yield inaccurate regions. This problem is known to be more challenging with histology images than with natural ones, since objects are less salient, structures have more variations, and foreground and background regions have stronger similarities. Therefore, computer vision methods for visual interpretation of CNNs may not directly apply. In this paper, a simple yet efficient method based on a composite loss is proposed to learn information from the fully negative samples (i.e., samples without positive regions), and thereby reduce false positives/negatives. Our new loss function contains two complementary terms: the first exploits positive evidence collected from the CNN classifier, while the second leverages the fully negative samples from training data. In particular, a pre-trained CNN is equipped with a decoder that allows refining the regions of interest. The CNN is exploited to collect both positive and negative evidence at the pixel level to train the decoder. Our method called NEGEV benefits from the fully negative samples that naturally occur in the data, without any additional supervision signals beyond image-class labels. Extensive experiments[1] show that our proposed method can substantial outperform related state-of-art methods on GlaS (public benchmark for colon cancer), and Camelyon16 (patch-based benchmark for breast cancer using three different backbones). Our results highlight the benefits of using both positive and negative evidence, the first obtained from a classifier, and the other naturally available in datasets.

**Keywords:** Deep Learning, Weakly Supervised Learning, Histology Images, Visual Interpretability, Positive / Negative Evidence, Class Activation Maps.

## 1. Introduction

The analysis of histology images remains the gold standard in the assessment of many pathologies such as breast (He et al., 2012; Gurcan et al., 2009; Veta et al., 2014), colon (Shapcott et al., 2019; Sirinukunwattana et al., 2015; Xu et al., 2020) and brain cancer (Ker

---

1. Our code is publicly available: https://github.com/sbelharbi/negev.

et al., 2019; Khalsa et al., 2020; Xu et al., 2017). Typically, pathologists perform such tasks on large histology images. To alleviate the workload of pathologists, computer-aided diagnosis (CAD) has been developed with the main aim to help them provide more reliable decision with less effort. CAD typically relies on computer vision and machine learning algorithms, and has been increasingly focused on deep learning (DL) models (Rony et al., 2019). In recent years, weakly-supervised learning methods for interpretation of CNN classifiers have attracted much attention (Campanella et al., 2019; Daisuke and Shumpei, 2018; Rony et al., 2019; Spanhol et al., 2016b,a; Sudharshan et al., 2019). Their main advantage is the ability to classify an input image, while providing visual interpretation of the classifier's decision. Often, such an interpretation comes in the form of a heat map or soft segmentation indicating the regions of interest (ROI) associated with the classifier prediction (Rony et al., 2019).

Training accurate DL models to provide reliable interpretations requires image-class and pixel-wise annotations on large image datasets. While acquiring image-class annotations can be manageable for human experts, producing the dense (pixel-wise) annotations such as a segmentation is extremely expensive and time-consuming. This annotation bottleneck is even more severe when processing histology images, where manual segmentation of large images is intractable.

Recently, weakly supervised learning (WSL) (Choe et al., 2020; Zhou, 2017) has emerged as a proxy aiming to reduce the cost and burden of fully annotating large datasets. Such methods exploit weak annotations such as global/coarse or ambiguous labels, and even more unlabeled samples in order to alleviate the need for dense labels. In segmentation tasks, weak supervisory signal comes under different forms making it more attractive to several practical scenarios. This includes scribbles (Lin et al., 2016; Tang et al., 2018a), points (Bearman et al., 2016), bounding boxes (Dai et al., 2015; Khoreva et al., 2017; Kervadec et al., 2020), global image statistics such as the target-region size (Kervadec et al., 2019b; Bateson et al., 2019; Jia et al., 2017; Kervadec et al., 2019a), and image-level labels (Belharbi et al., 2022; Kim et al., 2017; Pathak et al., 2015; Teh et al., 2016; Wei et al., 2017b). In this paper, we focus on image-class labels, where we are only given the image class for each image, such as cancerous or not. The goal is to jointly classify an input image, and accurately interpret the classifier's decision – i.e., locate the ROIs associated with the class prediction. This is achieved without any pixel-wise supervision or other cues/supervision signals, such as the size of the target regions (Kervadec et al., 2019b). Note that interpretability research has expanded significantly in the recent years, and has attracted broad interest within the computer vision (Bhatt et al., 2020; Fong et al., 2019; Fong and Vedaldi, 2017; Goh et al., 2020; Osman et al., 2020; Samek et al., 2020; Zhang et al., 2020c; Belharbi et al., 2021) and medical imaging communities (de La Torre et al., 2020; Gondal et al., 2017; González-Gonzalo et al., 2020; Taly et al., 2019; Quellec et al., 2017; Keel et al., 2019; Wang et al., 2017).

Class activation mapping (CAM) methods are currently driving state-of-the-art deep WSL methods (Rony et al., 2019; Zhang and Zhu, 2018; Zhou et al., 2016). Using image-class labels, a CNN can be trained to classify images. Typically, this yields spatial activation maps that typically provide strong activations in the regions associated with the image class. This makes these methods attractive for both interpretability and weakly-supervised segmentation of the ROIs. Existing methods could be divided into the following two main categories (Rony et al., 2019). (1) Bottom-up methods: They rely on the forward signal to locate the ROIs associated with the image class. This includes spatial pooling techniques

over activation maps (Durand et al., 2017; Oquab et al., 2015; Sun et al., 2016; Zhang et al., 2018a; Zhou et al., 2016), multiple-instance learning (Ilse et al., 2018) and attend-and-erase based methods (Kim et al., 2017; Li et al., 2018; Singh and Lee, 2017; Wei et al., 2017b). (2) Top-down methods: Inspired by human visual attention, these methods have gained much attention, and were designed initially as visual explanatory tools (Simonyan et al., 2014; Springenberg et al., 2015; Zeiler and Fergus, 2014). Both categories rely on the forward and backward signals to determine ROIs. Examples include special feedback layers (Cao et al., 2015) and back-propagation errors (Zhang et al., 2016). In addition, gradients are used to interrogate a CNN as to the ROIs of a specific class label (Fu et al., 2020; Lin et al., 2013; Pinheiro and Collobert, 2015).

Currently, WSL methods yield promising performance over natural images, where an object typically has a color distribution that is different from the background. This makes for salient objects that are relatively easy to locate. However, histology images provide additional challenges (Rony et al., 2019). In particular: (1) objects are less salient and difficult for non-experts to locate: Often, the ROIs have similar overall color/texture as the background. This requires an expert to decide what is relevant in the image with respect to the image class. (2) images are highly unstructured: Depending on several factors such as biopsy location/angle, images do not present repetitive global patterns that a machine learning model can learn. (3) high intra-class variability, due in large part to the high variation in structure, and the Hematoxylin and Eosin (H&E) staining process. This yields images from the same class that are completely different. This is challenging for a machine learning model as they often rely on the discovery of common shapes/pattern within a class.

Since most deep WSL methods were designed and evaluated on natural images, their direct application to histology images is limited by the above challenges. Without any pixel-level supervision, the task becomes more difficult. In paper, we consider using the uncertain knowledge learned by a classifier to guide a decoder towards better localization of ROIs. In addition to the positive evidence that indicates potential ROIs, we focus on negative evidence that points to potential background. This is useful with histology images since they often incorporate tissue (background) and cells (foreground). Moreover, we exploit fully negative samples, which are samples without any ROI, that can be easily obtained in practice. Such samples hold valuable information that indicates how a background should appear, and is expected to help a model distinguish between the foreground and background. Overall, we advocate using both positive and negative evidence to improve the ROI segmentation, thereby improving the interpretability aspect. Since our method is focused on exploiting NEGative EVidence, we name it *NEGEV*.

Our main contributions is summarized as follows. (1) To improve the interpretability of a classifier, we leverage negative evidence. This is expected to help our model discern the foreground from background more accurately. We propose a simple yet efficient method based on an original composite loss to learn information from the negative samples, while using only the image class as supervision. In contrast with existing works, which often uses the pixel-wise positive evidence collected from a classifier, we also explore the negative evidence, and integrate it with positive evidence, thereby taking advantage of the fully negative samples already available in the data. (2) Our training loss contains two complementary terms: One term exploits evidence collected from the classifier, and a second deals with the fully negative samples. When a sample is fully negative, knowledge from the classifier is not used. (3)

Following the experimental protocol in (Choe et al., 2020), we compare several WSL methods on two challenging histology benchmarks: GlaS colon cancer data set, and Camelyon16 patch-based benchmark for breast cancer. Our results show the benefits of using both negative and positive evidence, *i.e.*, the one obtained from a classifier and the one naturally available in datasets. We provide an ablation study to highlight the impact of each term.

## 2. Proposed method

### 2.1. Notation

Let $\mathbb{D} = \{(\boldsymbol{X}, y)_i\}_{i=1}^N$ denotes a training set, where $\boldsymbol{X} : \Omega \subset \mathbb{R}^2$ is an input image and $y \in \{1, \cdots, K\}$ is its global label, with $K$ the number of classes. $\mathbb{D}^- \subset \mathbb{D}$ denotes the set of all negative training samples that do not contain any ROI. The architecture of our model is similar to U-Net (Ronneberger et al., 2015). It contains two parts: (a) classification module $g$, and (b) segmentation module (decoder) $f$ to output two activation maps, one for the foreground and the other for background. The classifier $g$ is composed of an encoder backbone for building features, and a pooling head to yield classification scores. During the training phase of our method, all the weights of the classifier are frozen, and only the decoder weights $\boldsymbol{\theta}$ are updated.

The decoder generates softmax activation maps denoted as $\boldsymbol{S} = f(\boldsymbol{X}) \in [0,1]^{|\Omega| \times 2}$, where $\boldsymbol{S}^0, \boldsymbol{S}^1$ refer to the background and foreground maps, respectively. We note that the map $\boldsymbol{S}^1$ is class-agnostic. Let $\boldsymbol{S}_p \in [0,1]^2$ denotes a row of matrix $\boldsymbol{S}$, with index $p \in \Omega$ indicating a point within $\Omega$. Using the pre-trained classifier $g$, we collect positive and negative evidence at the pixel level using the normalized class activation map (CAM) corresponding to the true class $y$ of the input sample. We refer to this CAM as $\boldsymbol{C} \in [0,1]^{|\Omega|}$.

### 2.2. Sampling positive/negative evidence

We trained the decoder using the collected positive/negative evidence from CAM $\boldsymbol{C}$ and the negative-evidence subset $\mathbb{D}^-$. To sample pixels from $\boldsymbol{C}$, we rely on the CAM magnitude at each pixel. We assume that high activations indicate a potential foreground (ROI), while low activations indicate background, *i.e.*, absence of ROI.

For a training sample, we define two stochastically sampled subsets of pixels, $\mathbb{C}^+$ and $\mathbb{C}^-$, as foreground and background regions, respectively, estimated as follows:

$$\mathbb{C}^+ = \psi(\boldsymbol{C}), \quad \mathbb{C}^- = \psi(1 - \boldsymbol{C}), \tag{1}$$

where $\psi(\boldsymbol{C})$ is a *pixel* sampling function based on the multinomial distribution. For the foreground regions, we use the CAM activations as sampling weights, thereby encouraging the sampling of pixels with high activations. However, to sample background regions, we use the inverse magnitude, *i.e.* $1 - \boldsymbol{C}$. For the foreground, we assign the pseudo-label 1 to pixels in $\mathbb{C}^+$ whereas, for the background, pixels in $\mathbb{C}^-$ are associated with label 0. An unknown label is assigned to the pixels that were not sampled. Let $Y$ denotes the *partially* pseudo-labeled mask for sample $\boldsymbol{X}$, where $Y_p \in \{0,1\}^2$, with the labels being equal to 0 for the background, and to 1 for the foreground.

Due to the potential label noise in the collected pseudo-labels $\mathbb{C}^+$ and $\mathbb{C}^-$, we randomly sample only a few pixels, $n$, for each region, independently for each mini-batch gradient

update, and training image. Conceptually, this simulates extreme dropout (Singh and Lee, 2017; Srivastava et al., 2014), where a large part of a signal is dropped out, and only a small portion of it is considered[2]. However, while dropout is performed with uniform sampling, our sampling is weighted with importance. Intuitively, this prevents the model from quickly fitting the wrong labels, and gives more time for consistent ROIs to emerge during training.

Learning using the randomly estimated pseudo-annotation is achieved through partial cross-entropy. Let $\boldsymbol{H}(Y_p, \boldsymbol{S}_p) = -(1 - Y_p) \log(1 - \boldsymbol{S}_p^0) - Y_p \log(\boldsymbol{S}_p^1)$ denotes the standard binary cross-entropy between $\boldsymbol{S}_p$ and pseudo-label mask $Y_p$ at pixel $p$. This partial training loss over one sample reads:

$$\min_{\boldsymbol{\theta}} \quad \sum_{p \in \{\mathbb{C}^+ \cup \mathbb{C}^-\}} \boldsymbol{H}(Y_p, \boldsymbol{S}_p) \ . \tag{2}$$

Minimizing Eq. (2) enables the model to approximately discriminates between the foreground and background regions.

### 2.3. Accounting for fully negative samples

In several real applications, we can easily access fully negative samples. While such samples do not indicate any ROI, they yield a valuable information about what is not a ROI. We consider using these samples $\mathbb{D}^-$ for learning at the pixel level. This is expected to provide the model with useful information that helps in discriminating the foreground and background regions. To indicate that there are no ROIs in fully negative samples, we simply add a cross-entropy term $\boldsymbol{H}(Y_p, \boldsymbol{S}_p)$ that encourages predicting all pixels as background. We formulate this as:

$$\min_{\boldsymbol{\theta}} \quad \sum_{p \in \Omega} - \log(1 - \boldsymbol{S}_p^0) \ , \forall \boldsymbol{X} \in \mathbb{D}^- \ . \tag{3}$$

### 2.4. Overall training loss

The full training loss consists of the two exclusive terms defined above. The first term is defined over partially annotated regions, and a second term is defined over fully negative samples. Over a single training sample $\boldsymbol{X}$, the full loss is formulated as,

$$\min_{\boldsymbol{\theta}} \quad \mathbb{1}_{\boldsymbol{X} \in \mathbb{D}^-} \left( \sum_{p \in \Omega} - \log(1 - \boldsymbol{S}_p^0) \right) + (1 - \mathbb{1}_{\boldsymbol{X} \in \mathbb{D}^-}) \left( \lambda \sum_{p \in \{\mathbb{C}^+ \cup \mathbb{C}^-\}} \boldsymbol{H}(Y_p, \boldsymbol{S}_p) \right) \ . \tag{4}$$

where $\mathbb{1}_{\boldsymbol{X} \in \mathbb{D}^-}$ is the indicator function, and $\lambda$ is a weighting coefficient to account for the noise in the labels.

## 3. Experiments

Since we aim to evaluation segmentation of ROI of a classifier, image-class and pixel-wise annotation are required. There are two public histology datasets with both annotation (Rony et al., 2019). Image-class is used for training and evaluation of classification task. Pixel-wise labels are exclusively used for evaluation.

---

2. In all our experiments, for each mini-batch based stochastic gradient descent update during training, we randomly sample one ($n = 1$) single pixel per region, and such a sampling is done independently for each training image.

### 3.1. Datasets

**(a) GlaS dataset (Sirinukunwattana et al., 2015) (`GlaS`):** It is a histology dataset for colon cancer diagnosis[3]. It contains 165 images from 16 Hematoxylin and Eosin (H&E) histology sections and their corresponding labels. For each image, both pixel-level and image-level annotations for cancer grading (i.e., benign or malign) are provided. The whole dataset is split into training (67 samples), validation (18 samples) and testing (80 samples) subsets, as in (Rony et al., 2019). **(b) Camelyon16 patch-based benchmark (Rony et al., 2019) (`CAMELYON16`):** This benchmark is derived from the Camelyon16 dataset (Ehteshami Bejnordi et al., 2017), which contains 399 whole-slide images (normal or metastatic) for detection of metastases in H&E stained tissue sections of sentinel auxiliary lymph nodes (SNLs) of women with breast cancer. In (Rony et al., 2019), the authors designed a protocol to sample images with global and pixel-level annotations. Following this protocol, a patch can either be *(i)* normal without any metastatic regions, *(ii)* or metastatic with both normal and metastatic or only metastatic regions. In this work, we consider the benchmark containing images of size 512x512, and use the same split as in (Rony et al., 2019). This benchmark contains a total of 48,870 samples: 24,348 samples for training, 8,858 samples for validation, and 15,664 samples for testing. All subset have balanced global labels. Among both datasets, only `CAMELYON16` dataset contains fully negative samples. For `GlaS` datset, $\mathbb{D}^- = \varnothing$.

### 3.2. Experimental Protocol

We follow the same protocol in (Choe et al., 2020). Authors introduced a well defined setup to evaluate ROI obtained by weakly supervised classifier. This protocol entails two main elements: model selection, and evaluation metric at pixel level. Authors in (Choe et al., 2020) also define the area under the pixel precision and recall curve as an evaluation metric at the pixel-level. Since WSL method rely on thresholding, such metric marginalizes the threshold value. The metric is named `PxAP` where the higher the value is the better. We use the same set of thresholds in the interval $[0, 1]$ with a step of 0.001. Model selection is another critical issue. Classification and segmentation task are shown to be antagonist tasks (Choe et al., 2020). While segmentation task converges at the very early training epochs, classification task converges at the last epochs. Therefore, to yield better segmentation of ROI, an adequate model selection protocol is required. Following (Choe et al., 2020), we randomly select few samples in validation set where we have access to their segmentation. Such samples are exclusively used for early stopping using `PxAP` metric. Over `GlaS` dataset, we randomly select 3 samples per class, *i.e.* 6 samples in total. For `CAMELYON16`, we select 5 samples per class, *i.e.* 10 samples in total. For image-class evaluation, we use standard classification accuracy (`CL`).

### 3.3. Implementation details

The training of all methods is performed using SGD with 32 batch size (Choe et al., 2020), 1000 epochs for `GlaS`, and 20 epochs for `CAMELYON16` (Rony et al., 2019). We use weigh decay of $10^{-4}$. Images are resized to 256x256, and patches of size 224x224 are randomly sampled for training. Since all the methods were evaluated on natural images, we can not

---

3. The Gland Segmentation in Colon Histology Contest: https://warwick.ac.uk/fac/sci/dcs/research/tia/glascontest

use the reported best hyper-parameters in the original papers. For each method, including ours, we perform a search of the best hyper-parameter over the validation set, including the learning rate. For each method, the number of hyper-parameters to tune ranges from one to six. We use three different common backbones (Choe et al., 2020), VGG16 (Simonyan and Zisserman, 2015), InceptionV3 (Szegedy et al., 2016), and ResNet50 (He et al., 2016). Since our method is dependent on a pretrained classifier, we choose a simple CAM-based method that is CAM method (Zhou et al., 2016) which has an average `PxAP` performance. We use U-Net (Ronneberger et al., 2015) with full pixel-annotation to yield an upper-bound segmentation performance.

### 3.4. Results

Image classification accuracy is reported in appendix. Tab.1 presents the segmentation performance. Overall, we note that `CAMELYON16` dataset is much more difficult than `GlaS`. In addition, there is a performance discrepancy between the different backbones. Over `GlaS`, Grad-based methods show interesting results compared to bottom-up methods. However, over `CAMELYON16` dataset, most methods yield poor $PxAP \in [29\%, 45\%]$. Only Deep MIL obtained 54%. This is due to the difficulty of this dataset. Using CAM method (Zhou et al., 2016) as a pretrained classifier, our method allows to improve the segmentation performance over both datasets. For example, simply using one random pixel for foreground/background (at each SGD step) collected from the classifier, we were able to improve `PxAP` of Inception from 50.5% to 70.1%. Over the three backbones, we improve the average `PxAP` of CAM method from 61.1% to 77.8% over `GlaS`, and from 33.8% to 58.9% over `CAMELYON16`. In addition, our method yielded consistent improvement independent from the backbone/dataset. Moreover, we obtained better results than most state-of-the-art methods. However, there is still a large performance gap between WSL methods and fully supervised method. The ablation study in Tab.2 shows the impact of each term of our loss. Over `GlaS`, using positive evidence helps improving the performance. Using negative evidence adds further improvements, especially when the positive evidence is not enough such as in Inception. Similar behavior is observed over `CAMELYON16`. However, using fully negative samples adds a large improvement. For instance, over VGG, the `PxAP` goes from 38.1% with evidence from the classifier, to 70.3% with fully negative samples.

### 3.5. Discussions and conclusion

We have shown that simply exploiting negative evidence bring additional improvement to the segmentation of ROI of a pretrained classifier. Generally, this allows to enhance the interpretability of any classifier that yields CAMs. A main limitation to our method is its dependence to the classifier. The quality of the collected evidence depends on the segmentation performance of the classifier. When this evidence is relatively good such as in `GlaS`, exploiting this information can easily bring large improvement. However, when dealing with very noisy evidence, the improvement is small such as in `CAMELYON16`. This brings us to a common issue in the literature that is learning with noisy labels which is a growing field (Song et al., 2020). A direction to improve upon this work is to minimize the dependency of the learning over the classifier evidence. One could filter out poor evidence using strong uncertainty measure (Gal, 2016). Local and global constraints, such as color/texture and

| | GlaS | | | | CAMELYON16 | | | |
|---|---|---|---|---|---|---|---|---|
| | VGG | Inception | ResNet | Mean | VGG | Inception | ResNet | Mean |
| **Weakly-Supervised Leaning Methods** | | | | | | | | |
| GAP (Lin et al., 2013) *(corr)* | 58.5 | 57.5 | 56.2 | 57.4 | 37.5 | 24.6 | 43.7 | 35.2 |
| MAX-POOL (Oquab et al., 2015) *(cvpr)* | 58.5 | 57.1 | 46.2 | 53.9 | 42.1 | 40.9 | 20.2 | 34.4 |
| LSE (Sun et al., 2016) *(cvpr)* | 63.9 | 62.8 | 59.1 | 61.9 | **63.1** | 29.0 | 42.1 | 44.7 |
| CAM (Zhou et al., 2016) *(cvpr)* | 68.5 | 50.5 | 64.4 | 61.1 | 25.4 | 48.7 | 27.5 | 33.8 |
| HaS (Singh and Lee, 2017) *(iccv)* | 65.5 | 65.4 | 63.5 | 64.8 | 25.4 | 47.1 | 29.7 | 34.0 |
| GradCAM (Selvaraju et al., 2017) *(iccv)* | 75.7 | 56.9 | 70.0 | 67.5 | 40.2 | 34.4 | 29.1 | 34.5 |
| WILDCAT (Durand et al., 2017) *(cvpr)* | 56.1 | 54.9 | 60.1 | 57.0 | 44.4 | 31.4 | 31.0 | 35.6 |
| ACoL (Zhang et al., 2018a) *(cvpr)* | 63.7 | 58.2 | 54.2 | 58.7 | 31.3 | 39.3 | 31.3 | 33.9 |
| SPG (Zhang et al., 2018b) *(eccv)* | 63.6 | 58.3 | 51.4 | 57.7 | 45.4 | 24.5 | 22.6 | 30.8 |
| GradCAM++ (Chattopadhyay et al., 2018) *(wacv)* | **76.1** | 65.7 | 70.7 | 70.8 | 41.3 | 43.9 | 25.8 | 37.0 |
| Deep MIL (Ilse et al., 2018) *(icml)* | 66.6 | 61.8 | 64.7 | 64.3 | 53.8 | **51.1** | **57.9** | 54.2 |
| PRM (Zhou et al., 2018) *(cvpr)* | 59.8 | 53.1 | 62.3 | 58.4 | 46.0 | 41.7 | 23.2 | 36.9 |
| ADL (Choe and Shim, 2019) *(cvpr)* | 65.0 | 60.6 | 54.1 | 59.9 | 19.0 | 46.0 | 46.0 | 37.0 |
| CutMix (Yun et al., 2019) *(eccv)* | 59.9 | 50.4 | 56.7 | 55.6 | 56.4 | 44.9 | 20.7 | 40.6 |
| Smooth-GradCAM (Omeiza et al., 2019) *(corr)* | 71.3 | **67.6** | 75.5 | 71.4 | 35.1 | 31.6 | 25.1 | 30.6 |
| XGradCAM (Fu et al., 2020) *(bmvc)* | 73.7 | 66.4 | 62.6 | 67.5 | 40.2 | 33.0 | 24.4 | 32.5 |
| LayerCAM (Jiang et al., 2021) *(ieee)* | 67.8 | 66.1 | **70.9** | 68.2 | 34.1 | 25.0 | 29.1 | 29.4 |
| NEGEV (ours) | **81.3** | **70.1** | **82.0** | **77.8** | **70.3** | **53.8** | 52.6 | **58.9** |
| **Fully Supervised Learning Methods** | | | | | | | | |
| U-Net (Ronneberger et al., 2015) *(miccai)* | 96.8 | 95.4 | 96.4 | 96.2 | 83.0 | 82.2 | 83.6 | 82.9 |

Table 1: `PxAP` performance on `GlaS` and `CAMELYON16` test sets.

| | GlaS | | | | CAMELYON16 | | | |
|---|---|---|---|---|---|---|---|---|
| **Methods** | VGG | Inception | ResNet | Mean | VGG | Inception | ResNet | Mean |
| CAM (Zhou et al., 2016) | 68.5 | 50.5 | 64.4 | 61.1 | 25.4 | 48.7 | 27.5 | 33.8 |
| Ours + $\mathbb{C}^+$ | 81.3 | 53.3 | 81.3 | 71.9 | 38.1 | 36.5 | 30.8 | 35.1 |
| Ours + $\mathbb{C}^-$ | 81.3 | 52.9 | 81.3 | 71.8 | 42.1 | 27.1 | 38.0 | 35.7 |
| Ours + $\mathbb{C}^+ + \mathbb{C}^-$ | 81.3 | 70.1 | 82.0 | 77.8 | 38.1 | 35.3 | 30.2 | 34.5 |
| Ours + $\mathbb{C}^+ + \mathbb{C}^- + \mathbb{D}^-$ | – | – | – | – | 70.3 | 53.8 | 52.6 | 58.9 |
| Improvement | +12.8 | +19.6 | +17.6 | +16.6 | +44.9 | +5.0 | +25.1 | +25.0 |

Table 2: Ablation study ($n = 1$). `PxAP` performance over test set. Our method uses evidence from CAM (Zhou et al., 2016) pretrained model. $\mathbb{C}^+$: positive evidence. $\mathbb{C}^-$: negative evidence. $\mathbb{D}^-$: fully negative samples. Bottom line: improvement of our fully method compared to the baseline CAM.

object size, could be used with adaptation to histology images. We note that our method can be easily generalized to other type of medical images.

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

## Appendix A. Proposal

We present in this section an illustration of our proposal (Fig.1).

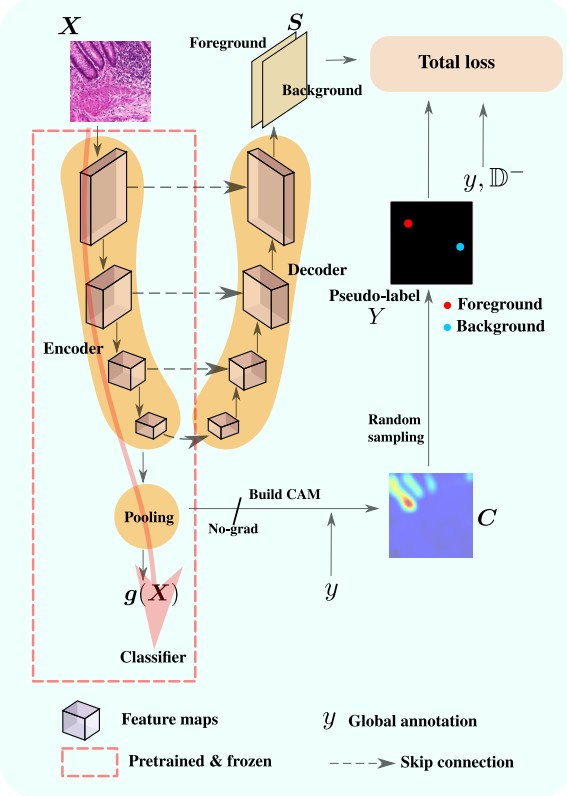

Figure 1: Our proposal. The model is composed of a pretrained classifier equipped with a decoder to yield refined ROI. During training, the classifier is frozen. Only the decoder is updated. The CAM $C$ produced by the classifier is used to sample positive/negative evidence to refine the prediction $S$. The total loss adjusts depending whether the sample is fully negative or not.

## Appendix B. Related work

Class activation mapping (CAM) was introduced in (Zhou et al., 2016). Authors show that spatial feature maps of standard deep classifier holds rich spatial information of ROI associated with the model decision. Since such model is trained only using image-class, CAMs tend to activate only on small discriminative regions while missing to cover the entire object. Most subsequent work on CAM come as an attempt to deal with this critical issue. Different extensions have been proposed. In particular, WSOL methods based on data enhancement (Choe and Shim, 2019; Singh and Lee, 2017; Wei et al., 2017a; Yun et al., 2019; Zhang et al., 2018a) aim to encourage the model to be less dependent on most discriminative regions and seek additional regions. For example, (Singh and Lee, 2017) divides the input

image into patches, and few of them are randomly selected during training. This forces the model to look for diverse discriminative regions. However, given this random information suppression in the input image, the CNN can easily confuse objects from the background because most discriminative regions were deleted. This leads to high false positives. Other methods consider improving the feature maps (Lee et al., 2019; Rahimi et al., 2020; Wei et al., 2021, 2018; Xue et al., 2019; Yang et al., 2020; Zhang et al., 2020a, 2018b). In (Wei et al., 2021), authors improve the features by considering shallow features of deep models.

Using negative evidence has been less explored in weakly supervised setup where the usage of positive evidence is more common (Zhou et al., 2016). Negative evidence has been used for instance in classification scores pooling. For instance, authors in (Durand et al., 2017) exploit negative evidence, along with positive evidence to compute class scores. Authors argue that such combination allows better regularization. However, it is not clear how it affects ROI. In contrast, other classification scoring functions model negative evidence to be later *excluded* from subsequent computations. For instance, max-pooling (Oquab et al., 2015) use the CAM activation magnitude to pick only high activations while ignoring low activations, *i.e.*, negative evidence. Other methods suppress negative evidence to prevent it from being considered. This is the case in deep multi-instance learning method (Ilse et al., 2018) where *attention* is used to pull a global representation of a bag using a weighted aggregation of its instance-level representations. Negative instances are expected to have low attention, hence, low contribution. Attention that neglects negative evidence has been also used in learning better spatial representations under weak-supervision (Choe and Shim, 2019).

Negative evidence has been exploited as well under the form of priors often in segmentation task. The aim is to suppress activations over negative regions. Such prior can be formulated as size constraint (Kervadec et al., 2019b,c; Pathak et al., 2015) where absent classes in an image are constrained to have zero size. This is also applied to reinforce the presence of background (Pathak et al., 2015). Authors in (**?**Belharbi et al., 2019) model the presence of background in an image using the response of a classifier. Over background regions, classifier is constrained to be the most uncertain in classification due to the lack of positive evidence for the corresponding class. In (Belharbi et al., 2022), authors use positive and negative evidence in addition to size priors and conditional random field (CRF) (Tang et al., 2018b). However, fully negative samples were not explored.

Authors in (Liu et al., 2021a) exploit positive and negative evidence but at patch level in order to refine segmentation. During training, a global and local models are trained to correctly classify image at coarse and local level, respectively. Positive patches are sampled around potential regions for foreground. However, negative patches are sampled from negative samples. Size constraints is used to prevent activation over entire image/patch. The patches are used to the local classifier to perform global classification. This is different from our work where sampled evidence is used directly to refine the ROI segmentation. The inference in their method requires both models and sampling patches to predict refined maps, which are remapped to build final segmentation.

**Relation to weakly supervised semantic segmentation (WSSS)**: Our work intersects with WSSS in terms of architecture and training. Due to the low resolution of standard CAMs, we considered to equip standard classifier with a decoder to yield refined ROI. This forms a model similar to standard segmentation models but it is able to classify an image and yield detailed ROI. Such decoder further more allows aligning ROI with external

pixel-wise cues when available in particular negative evidence as the decoder explicitly models background regions. This is impractical with standard CAMs due to their small size which is the size of the input image downscaled by factor up to 32.

In term of training, using pseudo-labels cues from a classifier is common in WSSS (Kolesnikov and Lampert, 2016; Liu et al., 2021b; Wang et al., 2020; Zhang et al., 2020b). To collect such pseudo-labels, typical WSSS rely on thresholding, or taking top activations. This can make them vulnerable to errors in fixed pseudo-labels. In contrast, our method follows a stochastic sampling of pixels. Such scheme has shown to be much more effective which highlights a limitation of using uncertain and constant pseudo-labels. In addition, the focus of our work is the usage of negative evidence that comes from CAMs or naturally occurs in datasets to mitigate a main drawback in standard CAM methods that tend to confuse foreground and background in histology images. While the goal of WSSS is only to segment an image, our method performance image classification and yield detailed segmentation of ROI. Moreover, the evaluation of WSSS is performed by taking the mask obtained over the softmax scores via `argmax` operation. Meanwhile, weakly supervised methods studied in this work threshold CAMs to yield ROI mask (Choe et al., 2020). The final metric marginalizes the threshold.

## Appendix C. Results

### C.1. Experimental details

To conduct our experiments, we first train a classifier $g$ over the training histology samples where we use standard backbones (VGG, Inception, ResNet50) following (Choe et al., 2020). Backbones weights are initialized using pre-trained weights from Image-Net (Krizhevsky et al., 2012). Once the classifier is trained, it is frozen including its backbone; and it is equipped with decoder as in a standard U-NET architecture (Fig.1). The decoder is randomly initialized, then trained using our proposed loss over the same training data.

**Patches**: To clarify our method:

- Our method operates on the entire image.

- The word 'patch' used in describing CAMELYON16 in Sec.3.1 should be 'image'. We used the same terminology as in(Rony et al., 2019), where they refer to an image as a patch because it is cropped from a whole slide image (WSI) – very high resolution (15000 x 15000) images of about 225 million pixels. They sampled sub-images of size 512x512 that are adequate for training machine learning models which we used in our experiments.

We have clarified this in the paper (see Section 3.1), where we refer to samples in CAMELYON16 as images. The rest does not change. During training, we resize the input image to 256x256 and crop random sub-images (patches) of size 224x224. During the test, we resize images to 224x224. This follows the protocol proposed in (Choe et al., 2020).

**Random sampling and number of labels**: In all our experiments, and unless mentioned differently, for each sample, and each SGD step, we randomly select one ($n = 1$) pixel for foreground, and one random pixel for background. Sampling is done independently for each sample. Therefore, the selected pixels for each sample will change at each SGD allowing exploring more regions using the provided evidence from the CAM activations. During the

entire training session of $e$ epochs, we have used a total of $2 \times e$ random pixels at most, per sample.

## C.2. Importance of randomness in pixels sampling

To show the importance of randomness in pixels selection, we conduct an addition experiment with static pixel selection, *i.e.*, without any randomness. In this experiment, we fix the selected pixel for each image. We select the same number of pixels $n$ per region as in randomness selection. However, instead of using multinomial random selection as proposed, we select top or low $n$ pixels. For foreground, we select top $n$ pixels ordered by activation magnitude. For the background, we select the low $n$ pixels. The results are presented in Tab.3 show the effectiveness of randomness. Nevertheless, even with static selection, our method still improves the base method CAM (Zhou et al., 2016). On one hand, random selection has the advantage over static selection by allowing exploring more regions under the guidance of CAMs' evidence. On the other hand, being able to generalize better indicates that randomness may be helping alleviating overfitting compared to static evidence where the target evidence is always the same.

| Methods | GlaS | | | | CAMELYON16 | | | |
| --- | --- | --- | --- | --- | --- | --- | --- | --- |
| | VGG | Inception | ResNet | Mean | VGG | Inception | ResNet | Mean |
| CAM (Zhou et al., 2016) | 68.5 | 50.5 | 64.4 | 61.1 | 25.4 | 48.7 | 27.5 | 33.8 |
| Ours ($n = 1$, random selection) | 81.3 | 70.1 | 82.0 | 77.8 | 70.3 | 53.8 | 52.6 | 58.9 |
| Ours ($n = 1$, static selection) | 77.7 | 60.3 | 76.5 | 71.5 | 57.5 | 47.4 | 42.8 | 49.2 |
| Performance drop | -3.6 | -9.8 | -5.5 | -6.3 | -12.8 | -6.4 | -9.8 | -9.6 |

Table 3: Ablation study ($n = 1$): **random** vs. **static** pixel selection. `PxAP` performance over test set. Our method uses evidence from CAM (Zhou et al., 2016) pretrained model. **Random sampling**: random $n$ pixels are selected at each SGD step, for each sample. Sampling is done independently for each image. **Static sampling**: same $n$ pixels are selected at each SGD step, for each sample (foreground: top $n$ pixels ordered by activation magnitude. background: low $n$ pixels ordered by activation magnitude.). Sampling is done independently for each image.

## C.3. Ablation study of $n$

We present in Tab.4 the impact the number of random sampled pixels per region, $n$, over the segmentation performance. Overall, varying the number of selected pixels does not break the performance of our method. Except Inception over `GlaS`, most models have seen slight degradation of performance when increasing the number of pixels. Given the noisy evidence, especially over `CAMELYON16`, using more pixels can be in disadvantage as the training may undergo fitting the wrong labels. It could be safer to consider sampling one random pixel to reduce the risk to pick wrong labels. Under the high uncertainty of the CAMs evidence, selecting large number of pixels increases the chance of *consistently* fitting wrong labels to the model through the entire training session.

| | GlaS | | | | CAMELYON16 | | | |
|---|---|---|---|---|---|---|---|---|
| $n$ | VGG | Inception | ResNet | Mean | VGG | Inception | ResNet | Mean |
| 1 | 81.3 | 70.1 | 82.0 | 77.8 | 70.3 | 53.8 | 52.6 | 58.9 |
| 2 | 81.3 | 52.9 | 81.3 | 71.8 | 69.7 | 51.1 | 47.2 | 56.0 |
| 3 | 81.3 | 52.9 | 81.3 | 71.8 | 69.7 | 51.9 | 47.2 | 56.2 |
| 4 | 81.3 | 55.0 | 81.3 | 72.5 | 69.7 | 50.0 | 47.2 | 56.6 |
| 5 | 81.3 | 52.9 | 81.3 | 71.8 | 69.7 | 53.4 | 47.2 | 56.7 |
| 10 | 81.3 | 53.7 | 81.3 | 72.1 | 69.7 | 52.6 | 47.2 | 56.5 |
| 20 | 81.3 | 52.9 | 82.2 | 72.1 | 69.7 | 51.3 | 47.2 | 56.0 |
| 50 | 81.3 | 52.0 | 81.3 | 71.5 | 69.7 | 53.8 | 50.3 | 57.9 |
| 100 | 81.3 | 53.4 | 81.3 | 72.0 | 69.7 | 50.5 | 47.2 | 55.8 |
| 500 | 81.3 | 52.9 | 81.3 | 71.8 | 69.7 | 51.5 | 47.6 | 56.2 |
| 1k | 81.3 | 53.7 | 81.3 | 72.1 | 69.7 | 51.2 | 48.5 | 56.4 |
| 2k | 81.3 | 53.0 | 81.3 | 71.8 | 69.7 | 51.5 | 47.2 | 56.1 |
| 3k | 81.3 | 54.2 | 81.3 | 72.2 | 69.7 | 50.4 | 48.5 | 56.2 |
| 4k | 81.3 | 52.9 | 81.3 | 71.8 | 69.7 | 52.9 | 47.2 | 56.6 |
| 5k | 81.3 | 53.2 | 82.7 | 72.4 | 69.7 | 51.4 | 47.7 | 56.2 |
| 10k | 81.3 | 52.9 | 81.3 | 71.8 | 69.7 | 52.1 | 47.2 | 56.3 |
| CAM (Zhou et al., 2016) | 68.5 | 50.5 | 64.4 | 61.1 | 25.4 | 48.7 | 27.5 | 33.8 |

Table 4: Ablation study of $n$, the number of random sampled pixels per region, in our method. `PxAP` performance over test set. Our method uses evidence from CAM (Zhou et al., 2016) pretrained model.

## C.4. Ablation study of $\lambda$

In our initial results, the value of $\lambda$ is empirically estimated using validation from values $\{1, 0.1\}$. We found that large values leads to better results and faster training. Low values are deemed to diminish the contribution of collected evidence into the gradient since it is the only source of learning. Typically, weighted terms are used in addition to other terms that can help boost gradients. However, in our case, collected evidence is the only training source. Under the same number of epochs, small value of $\lambda$ will slow learning since it multiplier coefficient of the learning rate. This slows learning, and requires more training epochs to converge to better solutions. We investigate large range values of $\lambda$. Results are presented in Tab.5 that show that low values lead to poor results. We recommend using large values for $\lambda$.

| | GlaS | | | | CAMELYON16 | | | |
|---|---|---|---|---|---|---|---|---|
| $\lambda$ | VGG | Inception | ResNet | Mean | VGG | Inception | ResNet | Mean |
| 1 | 81.3 | 70.1 | 82.0 | 77.8 | 70.3 | 53.8 | 52.6 | 58.9 |
| 0.1 | 81.3 | 50.8 | 81.3 | 71.1 | 69.7 | 52.5 | 47.1 | 56.4 |
| 0.01 | 80.3 | 52.9 | 73.0 | 68.7 | 69.5 | 50.6 | 51.0 | 57.0 |
| 0.001 | 80.2 | 52.9 | 56.8 | 63.3 | 65.3 | 51.2 | 38.1 | 51.5 |
| 0.0001 | 64.7 | 52.9 | 55.0 | 57.5 | 45.2 | 42.6 | 23.4 | 37.0 |
| CAM (Zhou et al., 2016) | 68.5 | 50.5 | 64.4 | 61.1 | 25.4 | 48.7 | 27.5 | 33.8 |

Table 5: Ablation study of $\lambda$ in our method. `PxAP` performance over test set. Our method uses evidence from CAM (Zhou et al., 2016) pretrained model.

## C.5. Visual results

We report few qualitative results of our method compared to the CAM method (Zhou et al., 2004) over both test datasets (Fig.2, 3). These results show the benefit of our method and how segmentation is affected when explicitly aligning pixels to a (noisy) target compared to CAMs (Zhou et al., 2004) where ROI emerge indirectly using global labels. Additional visual results over both datasets are presented in Fig,4, 5, 6, 7.

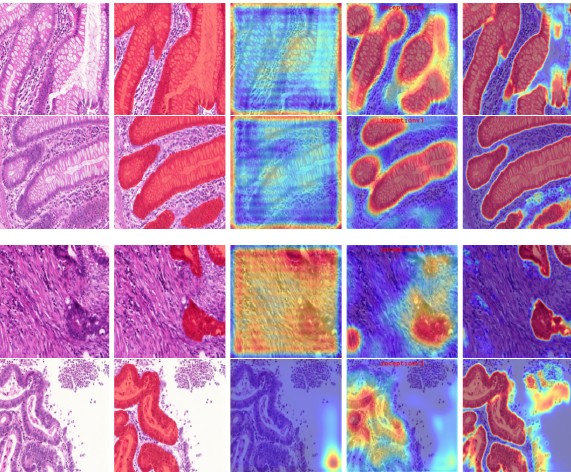

Figure 2: Predictions over test samples for `GlaS`. From left to right: input image, ground truth segmentation (ROI are indicated with red mask highlighting glands.), map from CAM method (Zhou et al., 2016), map from our proposed CAM, map from U-Net (Ronneberger et al., 2015) CAM. In all samples, strong CAM's activations indicated glands. Top two rows are benign. Bottom two rows are malignant.

## C.6. Classification performance

Tab.6 presents image classification accuracy (`CL`). As observed in (Choe et al., 2020), when using segmentation performance as a model selection metric, the classification performance is sub-optimal.

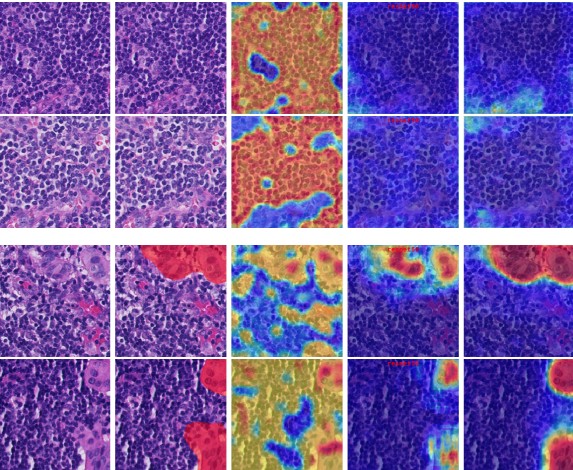

Figure 3: Predictions over test samples for `CAMELYON16`. From left to right: input image, ground truth segmentation (metastatic regions are indicated with red mask.), map from CAM method (Zhou et al., 2016), map from our proposed CAM, map from U-Net (Ronneberger et al., 2015) CAM. In all samples, CAM's activations indicate metastatic regions. Top two rows are normal. Bottom two rows are metastatic.

| | GlaS | | | | CAMELYON16 | | | |
|---|---|---|---|---|---|---|---|---|
| | VGG | Inception | ResNet | Mean | VGG | Inception | ResNet | Mean |
| **Weakly-Supervised Learning Methods** | | | | | | | | |
| GAP (Lin et al., 2013) *(corr)* | 46.2 | 93.7 | 87.5 | 75.8 | 50.0 | 50.0 | 68.1 | 56.0 |
| MAX-POOL (Oquab et al., 2015) *(cvpr)* | 97.5 | 86.2 | 46.2 | 76.6 | 50.0 | 82.0 | 71.4 | 67.8 |
| LSE (Sun et al., 2016) *(cvpr)* | 92.5 | 92.5 | 100 | 95.0 | 67.8 | 50.0 | 61.0 | 59.6 |
| CAM (Zhou et al., 2016) *(cvpr)* | 100 | 53.7 | 97.5 | 83.7 | 62.2 | 51.3 | 53.5 | 55.6 |
| HaS (Singh and Lee, 2017) *(iccv)* | 100 | 86.2 | 93.7 | 93.3 | 62.2 | 50.0 | 51.0 | 54.4 |
| GradCAM (Selvaraju et al., 2017) *(iccv)* | 97.5 | 85.0 | 98.7 | 93.7 | 40.2 | 34.4 | 29.1 | 34.5 |
| WILDCAT (Durand et al., 2017) *(cvpr)* | 55.0 | 86.2 | 96.2 | 79.1 | 50.0 | 50.0 | 50.0 | 50.0 |
| ACoL (Zhang et al., 2018a) *(cvpr)* | 100 | 95.0 | 46.2 | 80.4 | 50.0 | 50.0 | 50.0 | 50.0 |
| SPG (Zhang et al., 2018b) *(eccv)* | 53.7 | 53.7 | 72.5 | 59.9 | 65.1 | 50.0 | 49.4 | 54.8 |
| GradCAM++ (Chattopadhyay et al., 2018) *(wacv)* | 97.5 | 87.5 | 53.7 | 79.5 | 50.0 | 88.9 | 78.6 | 72.5 |
| Deep MIL (Ilse et al., 2018) *(icml)* | 96.2 | 81.2 | 98.7 | 92.0 | 86.6 | 71.3 | 88.1 | 82.0 |
| PRM (Zhou et al., 2018) *(cvpr)* | 96.2 | 53.7 | 96.2 | 82.0 | 50.0 | 75.5 | 50.0 | 58.5 |
| ADL (Choe and Shim, 2019) *(cvpr)* | 100 | 77.5 | 93.7 | 90.4 | 50.0 | 50.0 | 56.6 | 52.2 |
| CutMix (Yun et al., 2019) *(eccv)* | 100 | 86.2 | 100 | 95.4 | 66.8 | 80.8 | 53.0 | 66.8 |
| Smooth-GradCAM (Omeiza et al., 2019) *(corr)* | 100 | 97.5 | 97.5 | 98.3 | 50.0 | 88.5 | 51.0 | 63.1 |
| XGradCAM (Fu et al., 2020) *(bmvc)* | 100 | 91.2 | 88.7 | 93.3 | 82.1 | 88.9 | 82.3 | 84.4 |
| LayerCAM (Jiang et al., 2021) *(ieee)* | 100 | 90.0 | 53.7 | 81.2 | 85.8 | 47.4 | 82.1 | 71.7 |

Table 6: Image classification accuracy performance (`CL`) over `GlaS` and `CAMELYON16` test sets.

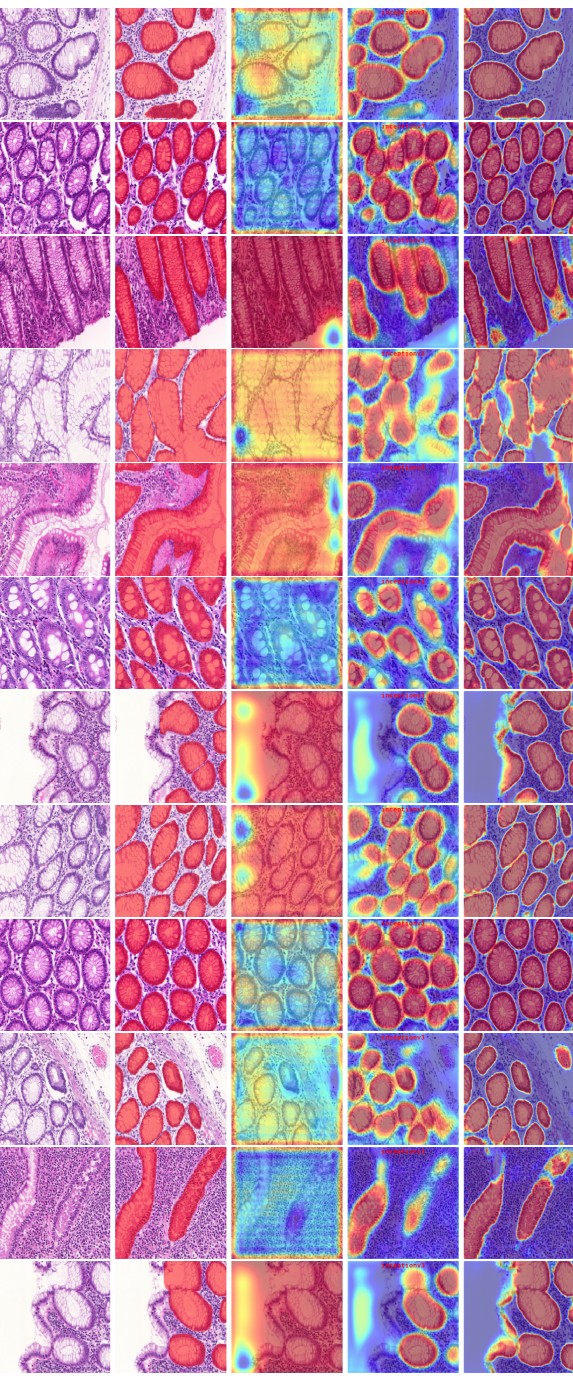

Figure 4: Predictions over benign test samples for `GlaS`. From left to right: input image, ground truth segmentation (ROI are indicated with red mask highlighting glands.), map from CAM method (Zhou et al., 2016), map from our porposed CAM, map from U-Net (Ronneberger et al., 2015) CAM. In all samples, strong CAM's activations indicated glands.

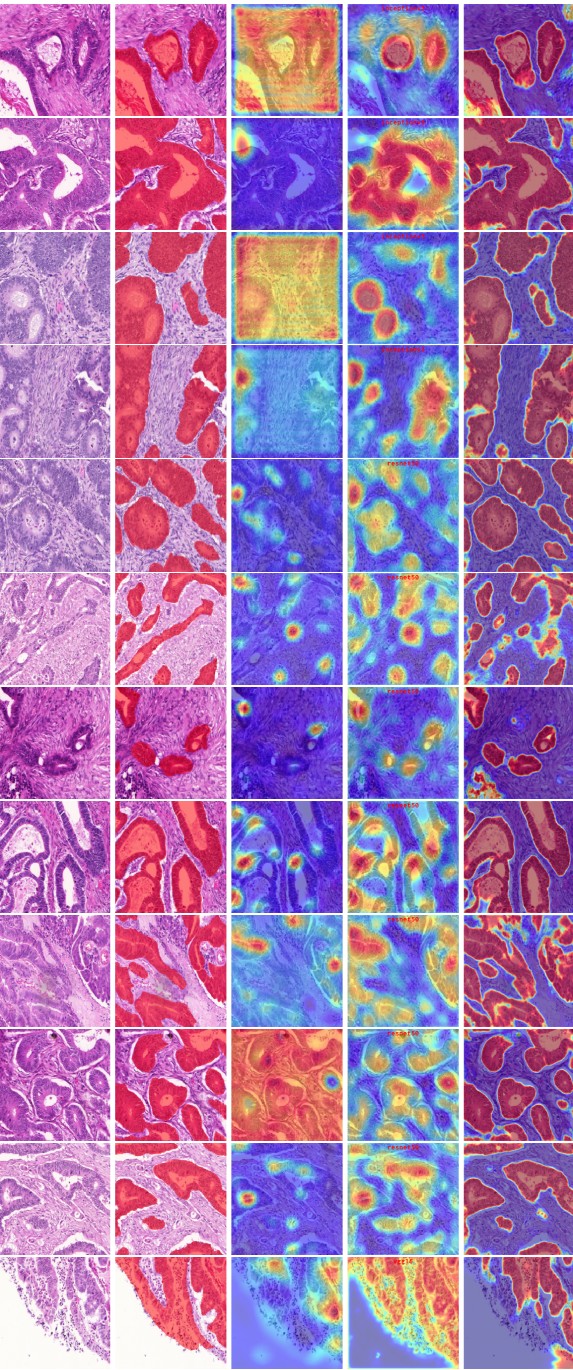

Figure 5: Predictions over malignant test samples for `GlaS`. From left to right: input image, ground truth segmentation (ROI are indicated with red mask highlighting glands.), map from CAM method (Zhou et al., 2016), map from our proposed CAM, map from U-Net (Ronneberger et al., 2015) CAM. In all samples, strong CAM's activations indicated glands.

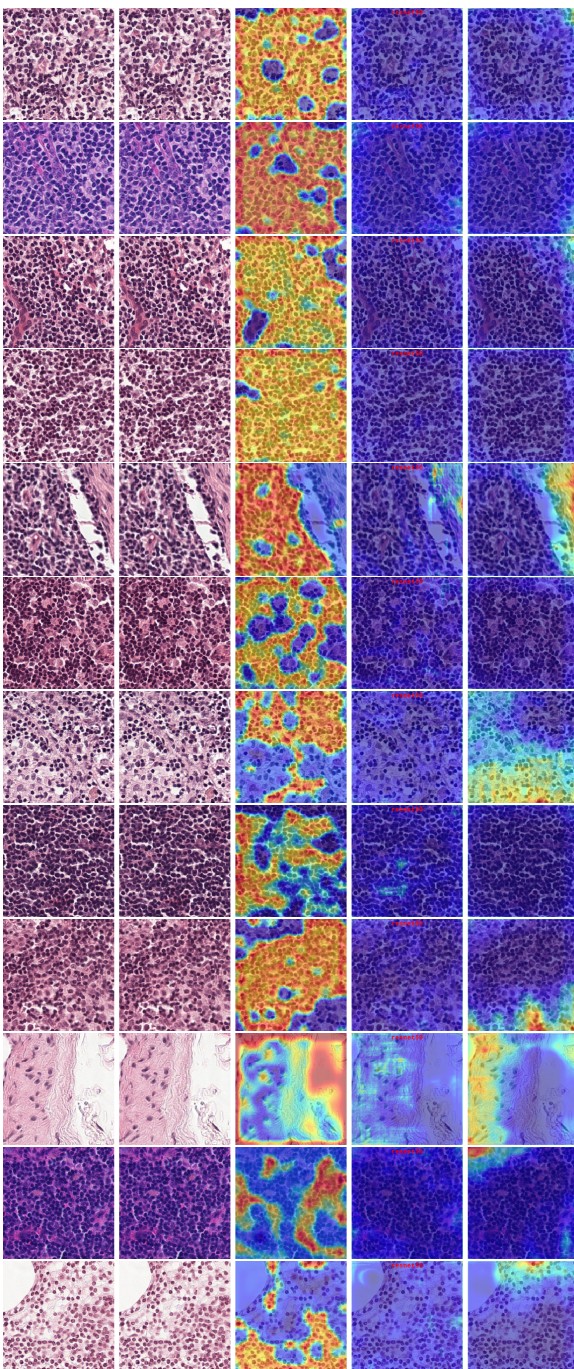

Figure 6: Predictions over normal test samples for `CAMELYON16`. From left to right: input image, ground truth segmentation (metastatic regions are indicated with red mask.), map from CAM method (Zhou et al., 2016), map from our proposed CAM, map from U-Net (Ronneberger et al., 2015) CAM. In all samples, CAM's activations indicate metastatic regions.

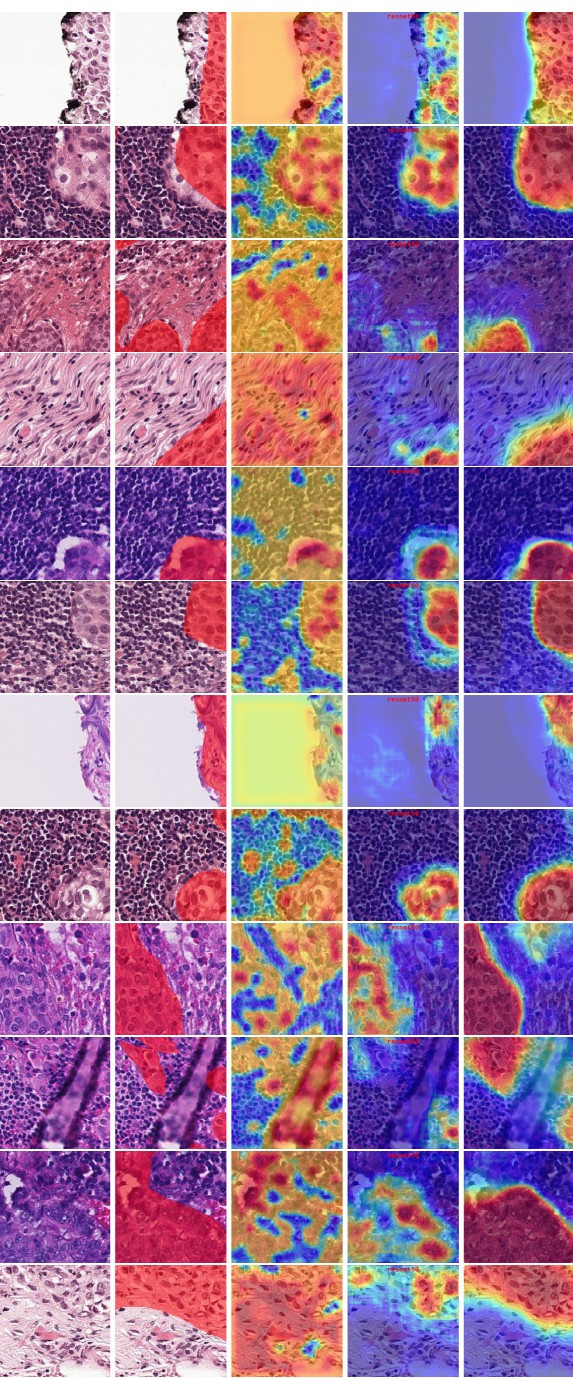

Figure 7: Predictions over metastatic test samples for CAMELYON16. From left to right: input image, ground truth segmentation (metastatic regions are indicated with red mask.), map of CAM method (Zhou et al., 2016), map from our proposed CAM, map from U-Net (Ronneberger et al., 2015) CAM. In all samples, CAM's activations indicate metastatic regions.

