# OpenReview forum: "Negative Evidence Matters in Interpretable Histology Image Classification"
_MIDL.io/2022/Conference — MIDL 2022_

### Official Review · Reviewer_vK7y · 2022-01-22

**Confidence:** 4
**Preliminary Rating:** 2
**Recommendation:** Oral, Poster

**Summary:**

The author proposes a way to utilize both the positive and negative evidence to segment the ROI in histology images with only global image-level labels. Based on sampling the positive evidence/negative evidence from CAM and additional negative evidence from fully negative samples, this paper trains a segmentation network to refine the CAM segmentation and achieves better segmentation performance. To this end, this paper reports more competitive results than the WSL methods.



**Strengths:**

- Illustrate the challenge in interpretation for classifiers of histology images.
- Achieving competitive performance than the WSL methods.
- Emphasize the importance of incorporating the information from fully negative samples for interpretation for classifiers of histology images.

**Weaknesses:**

The authors did not provide a structure figure to illustrate the network framework they are using.  Therefore, it is sometimes unclear to me:
- Is the segmentation module (decoder) operating on the whole image or just on the patch? From section 2.1, it seems that the decoder is operating on the whole image. However, in section 3.1, the author mentioned the patch size of Camelyon16 dataset and dividing image to patches; in section 3.3, the author mentioned "Images are resized to 256x256, and patches of size 224x224 are sampled for training." It is a little confusing to me. Yet I think a clearer description is importance to understand the core intuition of this paper


- "During the training phase of our method, all the weights of the classifier are frozen. Only the decoder weights, which θ, are updated." Do the decoder f and the classification module g share any module? (maybe the encoder)? I would speculate that they share the encoder, but this is not mentioned clearly in the paper (or I missed it). If they are sharing the encoder, this somehow means that both the g and f are operating on the image rather than patches. However, this would again make it quite confusing by mentioning the patches in the dataset section.


Some related work and context are missing:

- If the network is operating solely on the whole image level. Then, this pipeline would be very similar to the Weakly Supervised Semantic Segmentation  (WSSS) [1][2][3]:  1) a classification model is used to generate pixel-level annotations. This is often achieved by applying variations of Class Activation Maps combined with post-processing techniques. 2), these pixel-level annotations are used to train a segmentation model (UNet or Deeplab). Due to the higher resolution and refinement ability of the segmentation network, the result of WSSS would be better than the CAM (WSL). It would be therefore better if the author could give some context of the related work in WSSS and give some analysis of why an additional dedicated segmentation would be better than just using CAM from a classification network. I acknowledge that it is meaningful that the author introduces the WSSS pipeline to histology image analysis. However, this is not totally fair to just compare WSL results with WSSS.


- If the network decoder is working on top of the patchs, then this paper would be relevant to the pipeline in [4], where image-level CAM is provided as the initial guidance to train the patch-level network, in hopes of providing more fine-grained segmentation from the local network. Besides, that paper also mentions the importance of making use of the negative patches from negative samples, which is also a key component for this paper.



[1] Wang, Y., Zhang, J., Kan, M., Shan, S., & Chen, X. (2020). Self-supervised equivariant attention mechanism for weakly supervised semantic segmentation. In Proceedings of the IEEE/CVF Conference on Computer Vision and Pattern Recognition (pp. 12275-12284).

[2] Zhang, D., Zhang, H., Tang, J., Hua, X. S., & Sun, Q. (2020). Causal Intervention for Weakly-Supervised Semantic Segmentation. Advances in Neural Information Processing Systems, 33.

[3] Liu, S., Liu, K., Zhu, W., Shen, Y., & Fernandez-Granda, C. (2021). Adaptive Early-Learning Correction for Segmentation from Noisy Annotations. arXiv preprint arXiv:2110.03740.

[4] Liu, K., Shen, Y., Wu, N., Chłędowski, J. P., Fernandez-Granda, C., & Geras, K. J. (2021, February). Weakly-supervised High-resolution Segmentation of Mammography Images for Breast Cancer Diagnosis. In Medical Imaging with Deep Learning.


**Deanonymize Review:**

no

**Detailed Comments:**

Minor comments:

. some typos can be corrected in the updated version e.g. " thereby encouraging he sampling of pixels with high activations"

**Final Rating After The Rebuttal:**

4: Weak Accept

**Justification Of The Final Rating:**

Thanks for the response for the author, I am satisfied with the explannation. They add the figures to explain the network, which make it much easier to understand the setting of the framework. The additional section in the Appendix explains the connection between this methods w.r.t the related work sounds well. The additional ablation study is also interesting. Therefore, I decide to increase my rating to weak accpet.

**Paper Type:**

both

**Questions To Address In The Rebuttal:**

I would like the authors to clarify the concern I have about the network framework and training details (see *Weaknesses). Besides, I would like to hear from the authors the discussion about the connection and difference between this paper and the related work I mentioned ((see *Weaknesses). My initial rating is weak reject. However, I will consider increasing the score if the author could address my concerns.

**Special Issue:**

no

---

### Official Review · Reviewer_8aoE · 2022-01-23

**Confidence:** 4
**Preliminary Rating:** 4
**Recommendation:** Poster

**Summary:**

1. This paper proposes to explore negative evidence to improve the interpretability of a classifier, which may be helpful for a model to better distinguish back-/fore-ground.
2. Based on the above main idea, a loss is designed based on partially annotated regions and fully negative samples.
3. Experiments on two datasets show the effectiveness of the proposed method.
4. An open-source implementation is provided in this paper.

**Strengths:**

1. The paper is well-written and easy to follow.
2. The proposed method achieves state-of-the-art results.
3. The method is simple and effective.
4. The open-sour implementation is provided, which would be beneficial for the community to reproduce the results and develop related models.

**Weaknesses:**

One of the key contributions in this paper is enhanced interpretability. However, unlike other work like CAM/GradCAM, it seems that there is no visualization at all in this paper (though I can see some possible ones on their GitHub webpage). If p[ossible, it would be better to see the qualitative comparisons in the appendix. Since the experiments are conducted on mage datasets, it should be easy to do so.

**Deanonymize Review:**

no

**Paper Type:**

methodological development

**Questions To Address In The Rebuttal:**

1.  Please see "Weaknesses". My only concern is why the authors do not provide visualizations for the proposed interpretable method in the main body or appendix - I can only find something that could be visualizations on their webpage.
2. Further to my previous question, From Author Instructions  (https://2022.midl.io/author-instructions.html), it is not allowed to submit a long paper in parallel to other conferences. I am not sure whether this paper is also under review from other conferences because its arxiv version (https://arxiv.org/abs/2201.02445) looks quite different (much more detailed) from the current one.
3. If the paper is not the case of dual submission, it is worth an acceptance. If not, it seems to have violated the submission policy and I would change my score from "Weak accept" to "Strong Reject".

Looking forward to seeing the responses from the authors.

**Special Issue:**

no

---

### Official Review · Reviewer_95p1 · 2022-01-24

**Confidence:** 5
**Preliminary Rating:** 4
**Recommendation:** Poster

**Summary:**

This paper proposed a positive/negative evidence sampling approach to improving the segmentation performance of histology images. The ablation study showed the effectiveness of positive/negative evidence sampling and negative samples. State-of-the-art performance was achieved compared to other benchmarks.

**Strengths:**

1.	The authors implemented comprehensive comparison experiments and showed superior performance of the proposed method.
2.	The authors analyzed the specific challenges of histology images different from natural images and proposed the solutions accordingly.


**Weaknesses:**

1.	The title emphasized “negative evidence”, but in the ablation study, there seemed to be a decrease in performance when adding “EV-” onto “Ours + EV+”.
2.	The authors mentioned “simply using one pixel for foreground/background collected from the classifier” improved performance, but it is not clear how many pixels were used from the collected pseudo-labels in each experiment. Does this number change for different classifiers?
3.	The authors pointed out “the potential label noise in the collected pseudo-labels” such that a random sampling for each region was proposed. However, no experiment was performed to show the effectiveness of such subset sampling process.


**Deanonymize Review:**

no

**Paper Type:**

both

**Questions To Address In The Rebuttal:**

1.	Please add “ours + EV-” in the ablation study.
2.	Please clarify how the pixel quantity varies from the collected pseudo-labels.
3.	Please add experiments without random sampling from the collected regions.


**Special Issue:**

no

---

### Official Review · Reviewer_ZahS · 2022-01-25

**Confidence:** 4
**Preliminary Rating:** 4
**Recommendation:** Poster

**Summary:**

The authors aim to obtain soft segmentation (ROIs) in histology images which are more challenging than natural images. The authors propose a loss function that refines the ROIs of any pre-trained classifier by including negative samples in the loss function. Though most deep WSL methods have been designed and evaluated on natural images, directly applying these methods to histology images yields poor results as histology images are more challenging (ROIs have similar colors/texture as background etc). Thus, a loss function to refine the ROIs of any directly applied pre-trained classifier is proposed in this paper. The proposed method achieves better results than existing WSL methods on public benchmarks.


**Strengths:**

- The paper is well-written and adequately addresses prior works.
- It makes sense to make use of negative samples and propose a loss to improve the performances in a weakly-supervised setting.
- Extensive experiments have been conducted: 1) the proposed method outperforms existing WSL methods (Tab.1); 2) the proposed loss function has been adopted in 3 different baseline networks (VGG, Inception and ResNet as shown in Tab.2); 3) Ablation study has been carried out over different terms of the loss function (Tab.2).


**Weaknesses:**

- The paper claims that ‘Using negative evidence adds further improvements’, while this is not true from Tab.2. Specifically, for the CAMELYON16 dataset, EV+ + EV- option achieves an average of 34.5, which is inferior to using EV+ samples alone (35.1). Any reasons for this?
- For the cross entropy loss term (Eq.2), in the main text of the paper, it mentions that ‘we randomly sample only a few pixels for each region’; while in footnote 2 of page 5, it reminds ‘we randomly sample one single pixel per region’. Could you please make it more clear? And how would the number of sampling pixels affect the results?
- What’s the chosen value of the parameter lambda used in Eq.4? And any ablation studies on it?


**Deanonymize Review:**

no

**Detailed Comments:**

- In Table 2, the value under the ‘CAMELYON16 Mean’ column for the ‘CAM’ row is incorrect.
- A pictorial illustration of the framework (classifier + decoder + loss) should be provided to help with a quick understanding of the work.
- The pre-trained classifiers are VGG, InceptionV3, ResNet50. However, no details about the decoder have been provided.


**Final Rating After The Rebuttal:**

4: Weak Accept

**Justification Of The Final Rating:**

The rebuttal has addressed my concerns. The idea of the paper is well motivated and the empirical evaluation is solid, especially after the rebuttal period. Therefore I recommend this paper to be accepted.

**Paper Type:**

methodological development

**Questions To Address In The Rebuttal:**

I have listed my concerns in the weaknesses and detailed comments sections. These concerns are mainly regarding empirical results, explanations and illustrations. Hope the authors could address them in detail.

**Special Issue:**

no

---

### Meta-Review · Area_Chair_7UPR · 2022-02-17

**Recommendation:** Accept (Poster)
**Confidence:** 4

**Metareview:**

Although some further revision is required such as figures and details fo experiments, this paper highlight the benefits of leveraging both negative and positive evidence. Based on the consensus of reviewers and myself, a decision of accept is recommended.

---

### Decision · Program_Chairs · 2022-02-28

Accept